# How Microbes Affect Depression: Underlying Mechanisms via the Gut–Brain Axis and the Modulating Role of Probiotics

**DOI:** 10.3390/ijms23031172

**Published:** 2022-01-21

**Authors:** Kazunori Suda, Kazunori Matsuda

**Affiliations:** Yakult Central Institute, 5-11 Izumi, Kunitachi-shi, Tokyo 186-8650, Japan; kazunori-suda@yakult.co.jp

**Keywords:** gut microbiome, gut–brain axis, depression, probiotics

## Abstract

Accumulating evidence suggests that the gut microbiome influences the brain functions and psychological state of its host via the gut–brain axis, and gut dysbiosis has been linked to several mental illnesses, including major depressive disorder (MDD). Animal experiments have shown that a depletion of the gut microbiota leads to behavioral changes, and is associated with pathological changes, including abnormal stress response and impaired adult neurogenesis. Short-chain fatty acids such as butyrate are known to contribute to the up-regulation of brain-derived neurotrophic factor (BDNF), and gut dysbiosis causes decreased levels of BDNF, which could affect neuronal development and synaptic plasticity. Increased gut permeability causes an influx of gut microbial components such as lipopolysaccharides, and the resultant systemic inflammation may lead to neuroinflammation in the central nervous system. In light of the fact that gut microbial factors contribute to the initiation and exacerbation of depressive symptoms, this review summarizes the current understanding of the molecular mechanisms involved in MDD onset, and discusses the therapeutic potential of probiotics, including butyrate-producing bacteria, which can mediate the microbiota–gut–brain axis.

## 1. Introduction

Major depressive disorder (MDD) is a common but serious mental disorder characterized by symptoms such as depressed mood, anhedonia, fatigue, anxiety, irritability, insomnia, altered appetite, and suicidal ideation. It has been reported that around 280 million people worldwide suffer from MDD [1], and the results of a global burden of disease study revealed that, based on years lost to disability, depression ranks among the top 10 disabling diseases [2], indicating that MDD profoundly affects the individual’s quality of life worldwide. Although the etiologies underlying this disorder remain unclear, several hypotheses have been proposed to explain the underlying mechanisms of its pathogenesis. The monoamine hypothesis, which holds that depression arises from monoamine deficiency, has been the basis for the development of selective serotonin reuptake inhibitors (SSRIs) and serotonin noradrenaline reuptake inhibitors, widely used as antidepressants [3]. It has been observed that the increase in monoamines, which is the pharmacological effect of SSRIs, appears immediately after administration, whereas there is a latency in the response to the antidepressants for several weeks. The monoamine hypothesis is now considered to be part of the pathogenic mechanism of depression or its result [4]. It has become evident that multiple factors are intricately involved in the pathogenesis of depression, and disorders of the hypothalamus-pituitary-adrenal (HPA) axis involved in stress response, and neurogenesis/neuroplasticity dysfunction involving brain-derived neurotrophic factors (BDNF), have both been proposed as new hypotheses for the cause of depression.

Recent research has revealed that the gut microbiome influences human brain functions via the “gut–brain axis”, the bidirectional communication between the brain and the gut, which includes humoral and neural pathways. With the elucidation, in the psychiatric context, of the impact of the gut microbiota on human homeostasis via the gut–brain axis, the concept of a “microbiota–gut–brain (MGB) axis” has evolved [5]. Animal studies have revealed that gut dysbiosis is involved in abnormal stress response, reduced neurogenesis, and neuroinflammation in the host, all of which can be linked to the onset of MDD. Experiments involving germ-free mice that received fecal microbiota transplantation (FMT) from MDD patients strongly indicated that the gut microbiota contributed to the onset of depressive symptoms [6,7]. Alterations in the gut microbiota composition have been investigated in several human studies. Knudsen et al. conducted a systematic review of 17 studies analyzing the gut microbiota of MDD patients, and identified an increase in *Eggerthella*, *Atopobium*, and *Bifidobacterium* (all of the *Actinobacteria* phylum), and a decrease in *Faecalibacterium*, as common features in such patients [8]. *Faecalibacterium* is known as a main component of butyrate-producing bacteria in the gut, and butyrate has been reported to maintain BDNF levels and neurogenesis in the hippocampus, and improve depressive behavior. In the case of *Bifidobacterium*, several reports have indicated an opposite pattern to that found in [8]. Aizawa et al., for example, reported lower counts of *Bifidobacterium* and lactobacilli in the gut microbiota of patients with MDD compared with healthy controls [9], results supported by a number of previous studies in animal models [10,11,12]. *Bifidobacterium* and lactobacilli are commonly used as probiotics, which are gaining attention for their ability to maintain human physiological homeostasis in the gut environment and immune system, as well as psychological homeostasis to reduce the risk of developing depression [13].

This paper reviews recent findings on the possible mechanisms underlying the onset of MDD that can be modulated by the gut microbiome via the gut–brain axis, and discusses candidate interventions that may modulate the pathophysiology of depression, with a focus on the use of probiotics including lactobacilli, bifidobacteria, and butyrate-producing bacteria.

## 2. Pathogenesis of Depression Related to the MGB Axis

In this section, we describe possible mechanisms underlying the onset of MDD that can be mediated by the gut–brain axis, focusing on an abnormal stress response, decreased neurogenesis, and neural inflammation; and consider the possible involvement of the gut microbiome in such mechanisms.

### 2.1. Abnormal Stress Response

It has long been noted that life situations involving chronic stress increase the risk for MDD onset. The HPA axis is well recognized as a key pathway of stress response in mammals. Stress stimulus promotes the release of hypothalamic corticotropin-releasing hormone (CRH), and CRH induces the release of adrenocorticotropic hormone (ACTH) from the pituitary gland. ACTH is, in turn, secreted into blood circulation to stimulate glucocorticoid (e.g., cortisol in humans and corticosterone in rodents) secretion from the adrenal cortex. In addition to their effects on metabolism and the immune system, glucocorticoids regulate their own secretion by suppressing CRH secretion and HPA axis activation, which is known as negative feedback. It has been suggested that negative feedback of this kind is impaired in MDD patients. Carroll et al. conducted a dexamethasone (DEX; a type of glucocorticoid) test on melancholic MDD patients, and showed elevated plasma cortisol levels in the patients [14], while a later-developed method (DEX/CRH test) demonstrated that negative feedback was impaired in the patients [15]. As a result, blood cortisol levels are continuously high in MDD patients, and brain exposure to high levels of cortisol induces inflammation and reduction in BDNF levels, which are thought to underlie the pathophysiology of MDD [16]. Recently, cumulative HPA axis activities have been assessed based on hair cortisol levels, and the results suggest that stress and depressive symptoms are related to high cortisol levels [17,18]. Furthermore, in stress-induced depression model animals, the blood corticosterone concentration increased, and the brain BDNF level decreased, with resulting inflammation in the hippocampus and prefrontal cortex [19,20,21], while chronic administration of corticosterone to rodents induced a similar pathogenesis [22,23]. These results suggest that HPA axis-mediated dysregulation of stress response (i.e., HPA axis hyper-activation) contributes to the development and progress of depressive symptoms.

Accumulating evidence suggests an involvement of the MGB axis in the modulation of the host’s stress response. In Sudo et al.’s landmark study, germ-free mice showed an enhanced secretion of plasma ACTH and corticosterone, when compared with specific-pathogen-free (SPF) mice, under restraint stress [10], and it was later shown that modulation of the gut microbiota could affect the stress response, including the HPA axis [24,25]. Meanwhile, numerous studies, both animal and human, have been performed on the role of probiotics in stress resilience [26,27,28,29,30]. It has been shown that oral administration of *Lacticaseibacillus paracasei* strain Shirota (LcS; basonym, *Lactobacillus casei* strain Shirota) provoked activity of the gastric branch of the vagal afferent to suppress the stress-induced increase in blood corticosterone and the activation of CRH-positive neurons in the paraventricular nucleus of the hypothalamus in rats exposed to water-avoidance stress [31]. In human trials on healthy medical students under academic examination stress, this probiotic strain suppressed the stress-related increase of salivary cortisol and the onset of physical symptoms [31]. These results suggest that the sensory information on the gut microenvironment, including probiotics administration, is transmitted to the brain via vagal afferents, resulting in modulation of the stress response.

### 2.2. Decreased Neurogenesis and Its Association with BDNF

The hippocampus is a unique structure in which neurogenesis is maintained throughout life. Several animal studies suggested that decreased neurogenesis in the hippocampus resulted in a reduction in its volume, and BDNF was involved in these changes [32,33]. BDNF, which is produced by nerve cells, microglia, and astrocytes in broad regions of the brain, plays an important role in the survival, maintenance, differentiation, and synaptic plasticity of nerve cells. Measures of blood and plasma BDNF levels are considered to reflect brain-tissue BDNF levels [34], and decreased serum BDNF levels have been observed in MDD patients [35,36]. Structural imaging studies have shown that serum BDNF levels correlate with hippocampal volume, with lesser volume being observed in MDD patients than in healthy controls [37,38]. Furthermore, an animal study demonstrated that the administration of SSRIs increased hippocampal BDNF levels and neurogenesis, accompanied by a reduction in depressive behavior [39], a result confirmed in a human study, where antidepressant treatment recovered decreased levels of serum BDNF in MDD patients, along with improvement in related symptoms [40,41]. Santarelli et al. investigated the effect of the chronic SSRI, fluoxetine, on mice with neurogenesis suppressed by X-irradiation, and found that the mice were insensitive to behavioral effects of fluoxetine, indicating that the effects of chronic SSRIs may be mediated by the stimulation of neurogenesis in the hippocampus [42]. These results suggest that a reduction in BDNF impairs neurogenesis, resulting in MDD onset, and that antidepressants could exert a beneficial effect on MDD symptoms by recovering the brain BDNF level.

Several factors are known to affect the BDNF level, including exposure to excessive stress, inflammation, and aging. In addition to these factors, the gut microbiome has been reported to play an important role in controlling the BDNF level of the host. Since Sudo et al. showed that hippocampal BDNF levels decreased in germ-free mice compared with SPF mice [10], the relationship between BDNF and the gut microbiota has been investigated in numerous studies, using germ-free, antibiotic-treated, depression model, and fecal microbiota-transplanted mice [43,44,45,46,47], while impaired hippocampal neurogenesis has been reported in germ-free mice [48]. As an increase in the BDNF level correlates with an improvement in depressive symptoms [35,36], modulation of the gut microbiome, to increase the brain BDNF level and neurogenesis, may lead to an improvement in the symptoms. Gut microbiota-derived metabolites, especially short-chain fatty acids, have been confirmed to be key molecular mediators in the MGB axis, and butyrate is one of the bacteria-derived candidates that can link the gut microbiota with brain BDNF regulation. After absorption at the colon, butyrate is utilized by colonocyte for energy production, some of which then travels through the systemic circulation and reaches the brain through the blood–brain barrier [49,50,51]. In animal studies, butyrate was shown to accelerate BDNF expression in the hippocampus via inhibition of histone deacetylase [52], and to improve depressive behavior in stress-induced depression model mice [53]. As previously mentioned, several studies have revealed a lower level of butyrate-producing bacteria in the gut microbiota of MDD patients, and butyrate-producing *Faecalibacterium* and *Coprococcus* bacteria were positively associated with higher quality of life indicators in a cohort study of the Flemish Gut Flora Project [54]. These results consistently indicate that the pathophysiology of MDD could be modulated by gut microbiota-derived butyrate through the maintenance of BDNF expression.

### 2.3. Neuroinflammation

The last decade or so has seen a focus on the involvement of inflammation (neuroinflammation and systemic inflammation) in depressive symptoms, and increased levels of inflammatory markers have been observed in MDD patients. A meta-analysis conducted by Wang and Miller indicated higher levels of IL-6 and IL-8 in the cerebrospinal fluid (CSF) of MDD patients, while Goldsmith et al. observed higher levels of IL-6 and TNF-α in the blood of such patients [55,56]. It has also been shown that peripheral blood C-reactive protein (CRP) is elevated in patients with MDD, and more so in treatment-resistant patients [57]. Plasma CRP is proposed as a peripheral biomarker that reflects peripheral and central inflammation. Furthermore, Felger et al. have shown that plasma CRP strongly correlates with CSF CRP, and associates with CSF inflammatory markers related to depressive symptom severity [58]. In terms of neuroinflammation, it is known that microglia play a pivotal role in the production of inflammatory cytokines, and Nie et al. showed that repeated social defeat stress induced activation of microglia via the Toll-like receptor (TLR)-2 and TLR-4, up-regulation of IL-1α and TNF-α, and increased social avoidance and anxiety [59]. It is also known that microglia are involved in the pathogenesis of MDD induced by peripheral LPS-associated inflammation. Weng et al. reported that there was an increased number of microglia in the prefrontal cortex of mice intraperitoneally injected with *E. coli* derived-LPS, which was accompanied by an increase in depressive behavior [60]. They also found up-regulation of CD11b, IL-1β, IL-6, and TNF-α gene expression in the prefrontal cortex of the mice, and the changes were suppressed by SSRI. It has also been shown that astrocytes are involved in the pathogenesis of stress- and LPS-induced inflammation, accompanied by depressive symptoms [61,62].

Some hypotheses relate inflammation to the pathogenesis of MDD. One focuses on alteration of the tryptophan metabolic pathway. It is known that tryptophan is metabolized to both serotonin and kynurenine, with kynurenine being further metabolized to quinolinic acid, which has neurotoxicity. It has been reported that the tryptophan metabolic pathway to kynurenine and quinolinic acid is activated in MDD patients [63,64], and that IFN-γ up-regulates the kynurenine pathway while the serotonin pathway is down-regulated [65]. These results suggest that neuroinflammation causes the metabolic pathway shift from tryptophan toward kynurenine, and that quinolinic acid impairs the neurons, inducing depressive symptoms. Another hypothesis focuses on a dysfunction of the HPA axis and neurogenesis. An activated glucocorticoid receptor (GR) inhibits the expression and function of inflammatory cytokine, whereas IL-1α suppresses GR activation to induce glucocorticoid resistance (i.e., a reduced sensitivity to glucocorticoid) [66]. IL-1, IL-6, and TNF-α have been shown to stimulate the HPA axis and induce the release of glucocorticoid [67]. Thus, inflammation causes both glucocorticoid resistance and HPA axis activation, resulting in hyper-activation of the HPA axis. Several reports have suggested another pathway, in which systemic administration of IFN-α or IL-1β down-regulates BDNF expression, suppressing neurogenesis in the hippocampus, though the precise mechanism remains unclear [68,69].

Before the MGB axis gained research attention, extensive research sought to elucidate the relationship between the gut microbiota and inflammation and immune response, and numerous studies have shown that the gut microbiota directly affects the balance between pro- and anti-inflammatory responses in the gut [70,71,72,73]. Butyrate is also known to exert anti-inflammatory effects on immune cells. It has been shown that butyrate inhibits the activation of NF-κB and production of inflammatory cytokines in peripheral blood mononuclear cells, as well as the production of TNF-α in neutrophils [74,75], while Furusawa et al. showed that butyrate induced differentiation of the T-regulatory cells to ameliorate colitis [76]. Regarding the function of butyrate in brain regions, a recent study showed that butyrate supplementation improved behavioral abnormalities and modulated microglial homeostasis in high-fat diet-fed mice [77]. In terms of the relationship between MDD and inflammation, intestinal mucosal barrier function has been suggested to be involved in MDD onset. Intestinal luminal antigens/toxins such as LPS cause systemic inflammation and depressive symptoms when they translocate into blood circulation [60]. LPS and inflammatory cytokines also induce blood–brain barrier dysfunction [78,79,80]. Thus, it is suggested that intestinal barrier dysfunction causes an influx of intraluminal antigens/toxins, inflammatory cytokines, T-cells, and macrophages to the brain, inducing neuroinflammation via activation of microglia and astrocytes. This claim is supported by the observation that intestinal barrier dysfunction was accompanied by inflammation in MDD patients, and the degree of dysfunction correlated with the severity of the MDD symptoms [81]. An impairment of the intestinal barrier function is suggested to be a consequence of stress, inflammation, and gut dysbiosis; and these factors are thought to interact with each other to exacerbate the impairment.

### 2.4. Other MDD-Related Factors

In this section, we discuss other MDD-related factors, such as sleep disorders, metabolic disorders, and neurotransmitter dysregulation, which are considered to be closely associated with MDD pathophysiology, though the involvement of the gut microbiome is less certain in this respect.

#### 2.4.1. Sleep Disorders

Sleep disorders are associated with psychiatric disorders (including MDD), and are recognized as a risk factor for their development [82]. A meta-analysis conducted by Zhai et al. indicated that both short and long sleep duration were significantly associated with an increased risk of MDD in adults [83]. Recently, Liu et al. showed that insomnia mediated the association between perceived stress and depression but did not affect perceived stress directly [84]. These results suggest that insomnia could be a medical condition on its own, rather than a symptom of depression, and that improving sleep conditions could attenuate or prevent stress-induced depressive symptoms. It has been shown that the gut microbiome can affect sleep conditions. Ogawa et al., for example, showed that gut microbiota depletion due to antibiotic treatment resulted in an altered sleep/wake architecture, with changes in the intestinal balance of neurotransmitters [85]. Furthermore, a human study based on electroencephalogram activity revealed that probiotic treatment improved sleep conditions [86,87]. The hypothalamus is a potential target of the gut–brain axis, because gut hormones are involved in appetite regulation in the hypothalamus, and some probiotics are known to affect neuronal activity in there [31]. The hypothalamus works as a key center for regulating the sleep-wake cycle, and is also the region where the HPA axis initiates a series of reactions for the maintenance of homeostasis. Given such functional associations, it is reasonable to suppose that the gut microbiota or its metabolites influence the brain functions, via enteroendocrine cells or afferent neurons, to improve sleep conditions and thereby modulate depressive symptoms.

#### 2.4.2. Metabolic Disorders

MDD patients often have metabolic disorders, and patients with metabolic disorders also tend to have depressive symptoms [88,89,90]. In recent years, it has been noted that systemic chronic inflammation is associated with intestinal mucosal barrier dysfunction, and partly involved in the development of obesity and type 2 diabetes [90,91]. As discussed in Section 2.3, systemic inflammation is considered a pathogenesis of MDD, hence metabolic diseases and MDD may share pathological mechanism commonalities. Elevated levels of the gut hormones ghrelin and leptin have also been reported in MDD patients [92,93]. These hormones, secreted from gastric endocrine cells and adipocytes, respectively, function in appetite regulation. It has become clear that the hormones are also involved in sleep, cognition, the reward system, and stress response (HPA axis activity), influencing brain functions from the digestive tract [94,95,96,97,98,99]. However, the significance of their effects on brain functions remains to be determined, because there is conflicting evidence regarding their activity. Some studies suggest that ghrelin and leptin enhance HPA axis activity, but the opposite effect has also been reported [97,99,100,101,102]. In MDD patients who are comorbid with metabolic disorder, manipulating the gut microbiota may have beneficial effects on depressive symptoms through the modulation of these hormones.

#### 2.4.3. Dysregulation of Monoamines and Gamma-Aminobutyric Acid

Monoamines are known to be involved in mood, emotion, arousal, appetite, motivation, anxiety, and the reward system [103,104,105,106,107,108,109,110]; additionally, dysfunction in the monoamine nervous system may be involved in depressive symptoms such as increased anxiety, decreased motivation, and anhedonia. Lesser levels of gamma-aminobutyric acid (GABA), the primary inhibitory neurotransmitter, have been observed in the brains of MDD patients [111]. Mann et al. reported that the GABA level in CSF was lower in MDD patients compared with healthy controls, and the lower level was correlated with increasing severity of anxiety in the patients [112]. GABA neurons are broadly present in the brain, and participate in many functions, including anxiety, motivation, and the reward system [113,114,115], as well as playing an important role in ameliorating MDD symptoms [116].

It is well-known that more than 90% of serotonin is present in the enterochromaffin cells located in the gastrointestinal tract, and is released in response to stimuli from the gut lumen, to regulate gut motility and perception. Gut microbiota-derived metabolites or components induce the synthesis and release of serotonin [117,118,119], and increased serotonin synthesis in the gastrointestinal tract may cause decreased levels of tryptophan in blood circulation, with a resulting reduction in brain serotonin synthesis. Furthermore, monoamine levels in lumen were shown to be significantly lower in germ-free mice compared to SPF-microbiota transplanted mice [120,121]. Some bacteria are known to produce monoamine [122], and several strains of lactic acid bacteria have been identified as having the ability to produce GABA [123]. The significance and role of these gut microbiota-derived neuroactive metabolites are largely unknown, but given their function, they may play a pivotal role in the MGB axis. For example, if they are absorbed in the gut (or enter due to intestinal-barrier dysfunction), they can activate sensory nerves in peripheral and enteral nerve or endocrine cells, and thereby transmit stimulation to the brain. There is, as of yet, insufficient evidence to determine the relation between neuroactive metabolites from the gut microbiota and neurotransmitters in the host brain; further investigation is required.

## 3. Proposals for Improving the Pathophysiology of MDD via the MGB Axis

Several studies have shown that depressive symptoms improved in patients with irritable bowel syndrome (IBS), when treated with FMT [124,125]. FMT is considered to restore or reconstruct the gut microbiota, and thus contribute to the amelioration of gastrointestinal and neuropsychiatric symptoms, although confounding in bidirectional associations between IBS and MDD must be considered. In addition to the structural modulation of the gut microbiota, a therapeutic approach focusing on the characteristics of specific bacteria is now applied to psychiatric diseases, including MDD. The following sections outline the therapeutic potential of probiotics, including butyrate-producing bacteria, to ameliorate the pathophysiology of MDD via the MGB axis (Figure 1).

### 3.1. Ameliorating the Stress Response

Several human trials have demonstrated the function of probiotics in controlling anxiety and depression. Probiotics that confer such benefits are now called psychobiotics, which have been defined as “a live organism that, when ingested in adequate amounts, produces a health benefit in patients suffering from psychiatric illness” [126]. Mohammadi et al. demonstrated that the administration of probiotic yoghurt or capsules containing *Lactobacillus acidophilus* LA5 and *Bifidobacterium lactis* BB12 for six weeks improved mental health parameters of petrochemical workers, as measured by the General Health Questionnaire (GHQ) and the Depression Anxiety and Stress Scale (DASS) [127]. Pinto-Sanchez et al. investigated the effects of probiotic powder containing *Bifidobacterium longum* NCC3001 on anxiety and depression in patients with IBS [128], and found that more patients provided with probiotic powder showed a reduction in depression scores compared to those given the placebo, while a functional magnetic resonance imaging analysis showed that responses to negative emotional stimuli in multiple regions of the brain were more reduced in the probiotic than in the placebo group. Benton et al. conducted a randomized controlled trial in healthy subjects consisting mainly of older adults, to examine the effects of LcS on mood and cognitive function [129]. In a subgroup with a high depressive index at baseline, intervention subjects showed a significant improvement in depressive mood compared with subjects who received a placebo control. These results were supported by those from an open trial using the same probiotic strain in patients with MDD or bipolar disorder [130]. There, it was demonstrated that depressive symptoms and sleep quality were improved after LcS treatment, and the effect was associated with the gut microbiota composition, namely, abundance of *Actinobacteria*, including *Bifidobacterium*.

Recent findings indicate that some probiotic strains can ameliorate stress-induced physiological changes as well. LcS, for example, suppressed stress-induced increases in glucocorticoid levels in both a human academic stress model (salivary cortisol) and a rat water-avoidance stress model (plasma corticosterone) [31]. In a similar study using a human academic stress model, heat-killed *Lactobacillus gasseri* CP2305 ameliorated chronic-stress-associated responses, including increased salivary cortisol levels and increased expression of stress-responsive microRNAs [131]. These actions might be mediated by direct neural communication between the gut and brain. Intraduodenal injection of *Lactobacillus johnsonii* La1 activated the gastric vagal afferents and inhibited renal sympathetic nerve activity [132]. LcS has shown similar results [31], and its effect on CRH-induced sympathetic activation is suppressed by vagotomy [133]. Finally, the administration of *Levilactobacillus brevis* SBC8803 (basonym, *Lactobacillus brevis* SBC8803) promotes the secretion of serotonin from the small intestine of mice [134], which may induce an activation of the intestinal branch of the vagal afferent. These results suggest that some probiotic strains modulate stress-induced activation of the HPA axis, and the subsequent onset of depression, by acting on the neuroendocrine system.

### 3.2. Maintenance of BDNF Expression and Neurogenesis

Given that impaired neurogenesis accompanied by BDNF reduction is involved in the pathophysiology of MDD, the recovery of BDNF expression is considered to be a promising therapeutic approach for the maintenance of neurogenesis. Animal studies using a chronic stress model demonstrated that administration of butyrate-producing bacteria (*Clostridium butyricum* or *Faecalibacterium prausnitzii*) attenuated depressive behavior, with an increase in BDNF levels [135,136]. In a prospective open-label trial by Miyaoka et al., *C. butyricum* MIYAIRI 588 was effective against depressive symptoms in antidepressant-resistant MDD patients, when used in combination with antidepressants [137]. Although further studies, using a larger, double-blind, parallel-group design, are required to confirm these findings, the use of butyrate-producing bacteria will attract increasing attention as a promising therapeutic approach for depression. The supplementation of lactic acid bacteria or bifidobacteria offers an alternative means to up-regulate butyrate production by the gut microbiota, because these bacteria produce abundant amounts of lactate and/or acetate, which are then metabolized to butyrate by butyrate-producing bacteria [138]. In chronic stress-induced depressive mice, *Bifidobacterium longum* subsp. infantis E41 increased the BDNF level, with a resulting decrease in depressive behavior, but did not modulate the increased levels of serum corticosterone. Additionally, E41 increased the cecal butyrate concentration, which correlated with the BDNF level and depressive behavior [139]. There are also several other strains of lactic acid bacteria and bifidobacteria that were shown to improve depressive symptoms with an increase in butyrate [140,141].

Recently, Wei et al. investigated the effect of indole, a tryptophan metabolite of colonic microbiota, on hippocampal neurogenesis, and showed that the neurogenic effect was mediated by the aryl hydrocarbon receptor (AhR) [142]. In that study, however, another AhR ligand, kynurenine, did not induce neurogenesis, suggesting ligand specificity in the AhR-mediated regulation of neurogenesis. It is known that indole is further metabolized to indoxyl sulfate, and the latter is involved in the onset and exacerbation of chronic kidney disease [143], so the use of indole and indole-producing bacteria must be carefully evaluated for its safety.

### 3.3. Anti-Inflammatory Effect

Butyrate-producing bacteria are thought to exert beneficial effects on depressive symptoms in terms of their anti-inflammatory function. In an animal study, administration of *F. prausnitzii* to trinitrobenzene sulphonic acid-induced colitis mice reduced the severity of colitis, with increasing IL-10 secretion and decreasing TNF-α and IL-12 secretion in the colon [144]. There are still few human-study results concerning the effect of butyrate-producing bacteria on inflammatory diseases, but a meta-analysis revealed a negative association between the abundance of *F. prausnitzii* and IBD activity [145].

Purton et al. conducted a systematic review and meta-analysis of the effect of probiotics on the tryptophan-kynurenine pathway, which indicated that probiotics can modulate the serum kynurenine level and kynurenine/tryptophan ratio [146]. Further, *Lactiplantibacillus plantarum* 299v (basonym, *Lactobacillus plantarum* 299v) reduced the kynurenine concentration in the plasma of MDD patients, along with improvement in cognitive function, though the depressive symptoms were not improved [147]; and administration of *Lactobacillus helveticus* R0052 and *Bifidobacterium longum* R0175 improved depressive symptoms and reduced the kynurenine/tryptophan ratio [148]. Further studies are needed to elucidate what kind of molecules in the kynurenine metabolic pathway are involved in depressive symptoms, and how such molecules affect the symptoms.

Tightening the intestinal mucosal barrier also plays an important role in suppressing systemic inflammation and consequent neuroinflammation. Numerous results suggest that the gut microbiome modulate the intestinal barrier function [149,150,151]. In this context, butyrate has been shown to enhance the intestinal barrier function through maintenance of the expression of tight junction proteins including claudins, occludin, and zonula occludens protein-1 (ZO-1) [152], and an animal study has demonstrated that oral administration of *C. butyricum* or butyrate reduces intestinal injury associated with severe acute pancreatitis [153]. Some probiotics are reported to ameliorate intestinal barrier dysfunction via different mechanisms. Ait-Belgnaoui et al. showed that *Lactobacillus farciminis* administration prevented a stress-induced increase in colonic paracellular permeability, by inhibiting myosin light chain phosphorylation, which induces epithelial cell cytoskeleton contraction [154]. Another probiotic strain, *Lactobacillus rhamnosus* GG (LGG), prevented the hydrogen peroxide-induced disruption of barrier function, by maintaining the expression and localization of occludin and ZO-1, which form the tight junctions between epithelial cells [155]. Furthermore, a human study showed that ingestion of LGG reduced the gastric hyper-permeability induced by indometacin [156], while, in another human study, *Lactiplantibacillus plantarum* strain WCFS1 (basonym, *Lactobacillus plantarum* strain WCFS1) up-regulated the duodenal expression of occludin and ZO-1, and in vitro experiments suggested that this effect was mediated by TLR-2 [157]. Overall, these results suggest that butyrate-producing bacteria and some probiotics can ameliorate depressive symptoms through restoration of intestinal barrier dysfunction. Additionally, it has been shown that the probiotic bifidobacterial strains improve colonic permeability in obese adults when administered with prebiotics, galacto-oligosaccharides [158], and the results allow us to suppose that the synbiotic combinations of probiotics and prebiotics are also effective to comorbid depressive symptoms in metabolic disorders.

## 4. Future Perspectives

The gut microbiota, as well as its dysbiosis, both closely related to brain function via the MGB axis, are involved in the pathogenesis of MDD. Though many issues remain to be addressed before the numerous mechanisms involved are fully understood, accumulating evidence suggests that gut dysbiosis induces an excessive stress response, and major bacterial metabolites, specifically butyrate, affects BDNF expression in the brain. This review has mainly focused on the mechanisms underlying the effect of microbes on depression, but it is also important to understand how microbes are affected by changes in the host’s psychological and physiological state. A very recent study showed that exposure to chronic stress reduced the secretion of α-defensins, effector peptides of innate enteric immunity produced by Paneth cells in the small intestine, which resulted in gut dysbiosis and an impairment of intestinal metabolite homeostasis [159]. To extrapolate the mechanisms revealed in animal studies to therapeutic effects in humans, it would be necessary to gain a deeper understanding of the bidirectional communication between the gut microbiota and brain.

It is becoming clear that abnormal stress response, reduced neurogenesis and BDNF expression, and neuroinflammation are involved in the pathogenesis of MDD. BDNF is thought to also be associated with a variety of other psychiatric disorders [160]. Hippocampal neurogenesis steadily decreases with aging, but drops sharply in patients with Alzheimer’s disease (AD) [161]. Neuroinflammation is also recognized as one of the potential mechanisms mediating neurodegenerative disorders, including AD [162]. Clarifying the relationship between the pathogenesis of MDD and the gut microbiome would also lead to better understanding of other neuropsychiatric disorders.

Probiotics, the gut microbiota, and their metabolites, especially SCFAs such as butyrate, play an important role in maintaining host homeostasis. Some probiotic strains induce excitation of the vagal afferents (with the stimulus then transmitted to the brain), and suppress stress-associated neural activation. It is expected that the means by which host cellular sensors detect bacterial components and induce signal transduction will be further elucidated, enabling identification of the bacterial components that act on the host cells, and determination of the specific mechanisms involved in their activity. To understand the dynamics of bacterial metabolites in the human body is another important issue, and a further challenge lies in determining whether butyrate reaches the brain via the blood stream, to directly influence the brain functions, or if another, peripheral tissue acts as an intermediate. Through these efforts, future research is expected to further clarify the underlying mechanisms of the pathogenesis of MDD, and lead to the development of promising microbiome-based therapeutics.

## Figures and Tables

**Figure 1 ijms-23-01172-f001:**
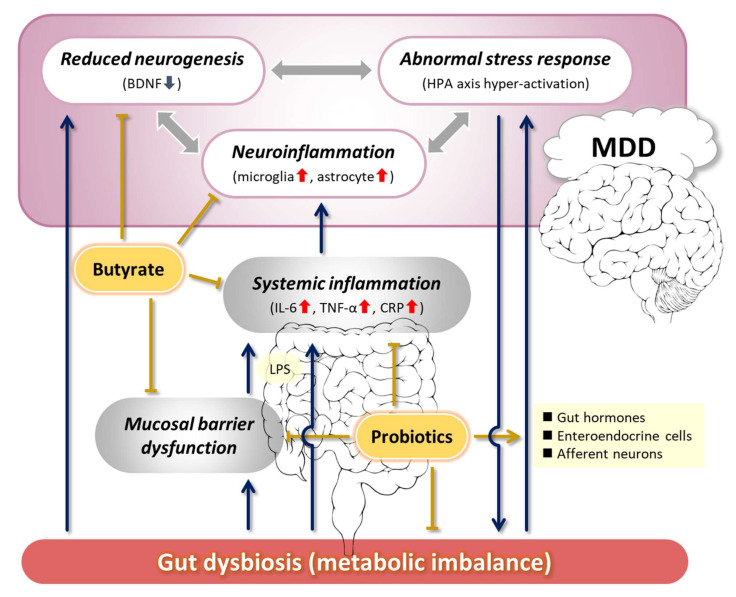
Underlying mechanisms for MDD via the gut–brain axis, and the modulating role of probiotics.

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
