# Peer review of "How Microbes Affect Depression: Underlying Mechanisms via the Gut–Brain Axis and the Modulating Role of Probiotics"

_ijms, 2022, doi:10.3390/ijms23031172_

Round 1

Reviewer 1 Report

Dear Kazunori Suda and Kazunori Matsuda, it was very interesting to review your paper, congratulations to this nice review. I have only few comments.

- Line 286/287: “…and increased serotonin synthesis in the gastrointestinal tract may result in its deficiency in the brain.” This hypothesis is very vague. To my knowledge, there is not really a connection between gut and brain serotonin levels.

- Figure 1: As you say in the text, high stress levels can induce gut dysbiosis. So I would suggest arrows pointing in both directions (not only from gut to stress).

- Line 303/304: “…that depressive symptoms improved in patients with irritable bowel syndrome (IBS), when treated with FMT…”. This is very vague. If patients are feeling better, you would of course expect less depressive symptoms. Is there proof of change in any of the mechanisms/metabolites/”biomarkers” that you elucidated in the sections before?

How do you define gut dysbiosis?

What is the advantage of taking probiotics compared to butyrate?

Author Response

Dear Kazunori Suda and Kazunori Matsuda, it was very interesting to review your paper, congratulations to this nice review. I have only few comments.

Response)
Thank you very much for taking time to review our manuscript. We are pleased to hear your very positive feedback.

Comment 1)
Line 286/287: “…and increased serotonin synthesis in the gastrointestinal tract may result in its deficiency in the brain.” This hypothesis is very vague. To my knowledge, there is not really a connection between gut and brain serotonin levels.

Response to Comment 1)
As over 90% of whole body’s serotonin is in gastrointestinal tract, we have made a hypothesis that increased serotonin synthesis in the gastrointestinal tract may cause decreased levels of tryptophan in blood circulation, with a resulting reduction in brain serotonin synthesis. We have modified the descriptions at the corresponding part in the revised version.

Comment 2)
Figure 1: As you say in the text, high stress levels can induce gut dysbiosis. So I would suggest arrows pointing in both directions (not only from gut to stress).

Response to Comment 2)
It’s just as you said. We added the arrows toward gut from stress in the figure.

Comment 3)
Line 303/304: “…that depressive symptoms improved in patients with irritable bowel syndrome (IBS), when treated with FMT…”. This is very vague. If patients are feeling better, you would of course expect less depressive symptoms. Is there proof of change in any of the mechanisms/metabolites/”biomarkers” that you elucidated in the sections before?

Response to Comment 3)
In the reports we cited in the manuscript (#124 and 125), no relevant biomarkers were investigated. As pointed out by the reviewer, confounding in bidirectional associations between IBS and MDD must be considered. We have added the descriptions at the corresponding part in the revised version.

Comment 4)
How do you define gut dysbiosis?

Response to Comment 4)
Generally, “gut dysbiosis” is defined as the condition of the gut microbiota with altered abundance, composition, and diversity when compared with the healthy controls. In the MDD patients, the compositional changes such as increased Actionobacteria and lowered Faecalibacterium were observed as mentioned in the manuscript.

Comment 5)
What is the advantage of taking probiotics compared to butyrate?

Response to Comment 5)
Butyrate-producing bacteria (probiotics) is expected to constantly produce butyrate in the gastrointestinal tract, which may increase a supply of butyrate to the brain than the transient increase by the butyrate administration. Regarding another role to enhance the gut barrier function, taking probiotics to produce butyrate in the gastrointestinal tract is thought to more effective to exert the role than taking butyrate that is being absorbed mainly in the upper part.

Reviewer 2 Report

The Review Article on the influence of microbes on depression analyzes the underlying mechanisms through the gut-brain axis and the modulating role of probiotics. The authors collected and analyzed a large amount of recent literature data, correctly systematized them in the main directions. I would like to emphasize the relevance of the topic. Due to the ongoing coronavirus pandemic, we are all under constant stress and at risk of mental complications, including major depressive disorders, sleep disturbances and anxiety. The proposed way of correcting problems, in my opinion, can make it possible to create a soft but effective therapy. It should be noted that the authors in their review paid much attention to the description of the molecular mechanisms of both pathologies and the action of various types of bacteria. I note that I was particularly interested in 2 topics of the review. First, a description of neurogenesis and its relationship to BDNF, including neuroinflammation. Second, a comparison of the effects of prebiotics and probiotics with conventional SSRI therapy is presented. The authors, while justifiably using the results of their own research, did not place special emphasis on them.

I believe that this review article is written concisely but informatively, meets the strict requirements of the journal and can be published without changes.

Author Response

The Review Article on the influence of microbes on depression analyzes the underlying mechanisms through the gut-brain axis and the modulating role of probiotics. The authors collected and analyzed a large amount of recent literature data, correctly systematized them in the main directions. I would like to emphasize the relevance of the topic. Due to the ongoing coronavirus pandemic, we are all under constant stress and at risk of mental complications, including major depressive disorders, sleep disturbances and anxiety. The proposed way of correcting problems, in my opinion, can make it possible to create a soft but effective therapy. It should be noted that the authors in their review paid much attention to the description of the molecular mechanisms of both pathologies and the action of various types of bacteria. I note that I was particularly interested in 2 topics of the review. First, a description of neurogenesis and its relationship to BDNF, including neuroinflammation. Second, a comparison of the effects of prebiotics and probiotics with conventional SSRI therapy is presented. The authors, while justifiably using the results of their own research, did not place special emphasis on them.

I believe that this review article is written concisely but informatively, meets the strict requirements of the journal and can be published without changes.

Response)
Thank you very much for taking time to review our manuscript. We are pleased to hear your very positive feedback.